# Reduction in Lens Epithelial Cell Senescence Burden through Dasatinib Plus Quercetin or Rapamycin Alleviates D-Galactose-Induced Cataract Progression

**DOI:** 10.3390/jfb14010006

**Published:** 2022-12-21

**Authors:** Yinhao Wang, Yulin Tseng, Keyu Chen, Xinglin Wang, Zebin Mao, Xuemin Li

**Affiliations:** 1Department of Ophthalmology, Peking University Third Hospital, Beijing 100191, China; 2Beijing Key Laboratory of Restoration of Damaged Ocular Nerves, Peking University Third Hospital, Beijing 100191, China; 3Department of Biochemistry and Molecular Biology, Health Science Center, Peking University, Beijing 100191, China

**Keywords:** cataract, senescence, senolytic, senescence-associated secretory phenotype

## Abstract

Senescent cells accumulate in aged organisms and promote the progression of age-related diseases including cataracts. Therefore, we aimed to study the therapeutic effects of senescence-targeting drugs on cataracts. In this study, a 28-day D-galactose-induced cataract rat model was used. The opacity index, a grading based on slit-lamp observations, was used to assess lens cloudiness. Furthermore, the average lens density (ALD), lens density standard deviation (LDSD), and maximum lens density (MLD) obtained from Scheimpflug images were used to assess lens transparency. Immunohistochemical stainings for p16 and γH2AX were used as hallmarks of senescence. We treated rat cataract models with the senolytic drug combination dasatinib plus quercetin (D+Q) and senescence-associated secretory phenotype (SASP) inhibitors. In comparison to control lenses, D-galactose-induced cataract lenses showed a higher opacity index, ALD, LDSD, and MLD values, as well as accumulation of senescent lens epithelial cells (LECs). After D+Q treatment, ALD, LDSD, and MLD values on day 21 were significantly lower than those of vehicle-treated model rats. The expression levels of p16 and γH2AX were also reduced after D+Q administration. In addition, the SASP inhibitor rapamycin decreased the opacity index, ALD, LDSD, and MLD values on day 21. In conclusion, D+Q alleviated D-galactose-induced cataract progression by reducing the senescent LEC burden in the early stage of cataract.

## 1. Introduction

Cellular senescence is a state of permanent cell cycle arrest that can occur in response to various stimuli [1]. One main feature of aged organisms is the accumulation of senescent cells, which promotes the occurrence of age-related and degenerative diseases [2].

Cataract is an age-related disease that is one of the leading global causes of blindness or moderate/severe vision impairment (MSVI) in those aged 50 years and older in 2020 (15.2 million cases [95% uncertainty intervals = 12.7–18.0] for blindness, 78.8 million cases [67.2–91.4] for MSVI) [3]. With increasing age, age-related changes occur in the lens, leading to lens aging and cataract formation [4]. Similar to other age-related diseases, cataracts have increased the numbers of senescent cells, namely lens epithelial cells (LECs) [5,6,7], and senescent LECs are associated with cataract progression. However, the exact functions of senescent LECs have rarely been investigated.

Age-related cataracts commonly cause nuclear and cortical opacities, while diabetes-related cataracts tend to cause opacities in the lens cortex. Compared with other cataract models, D-galactose-induced lens opacity possesses the characteristics of simplicity, feasibility, high success rate, and low mortality [8]. D-galactose induces cataracts by multiple underlying mechanisms, such as elevated intra-lens osmotic pressure, impaired cell-cell communications, and oxidative stress. [8,9]. It was reported that D-galactose was able to induce LEC senescence [10]. Therefore, D-galactose-induced cataracts were commonly used for the models of diabetic cataract and age-related cataract.

Reduction in the senescence burden significantly prolongs the health span of aged mice and alleviates aging and functional decline [11]. Thus, targeting senescent cells to delay aging and limit dysfunction, known as “senotherapy,” is gaining momentum. Senotherapy comprises two main strategies: senolytics and senomorphics. Senolytics are drugs that selectively kill senescent cells, whereas senomorphics are drugs that inhibit senescence-associated secretory phenotypes (SASPs). Senolytics were initially found through bioinformatic analyses targeting the senescent cell anti-apoptotic pathway (SCAP), including dasatinib (D), quercetin (Q), and fisetin [12]. D, a tyrosine kinase inhibitor approved for clinical antitumor therapy in 2006, is able to promote apoptosis by partially inhibiting Src kinase activity [13]. Q and F are natural flavonoids capable of inhibiting BCL-2 family members and other molecules of the SCAP network, acting to kill senescent cells [14]. The senolytic combination of D and Q has been proven to eliminate senescent cells and slow the progression of various age-related diseases, such as age-dependent intervertebral disc degeneration, intestinal senescence, and inflammation [15,16,17]. Senescent cells express proinflammatory cytokines, growth factors, chemokines, and proteinases, termed SASP, that promote the aging of organisms. mTOR is a serine/threonine kinase associated with a variety of metabolic and growth-promoting signaling pathways in cells [18]. The mTOR inhibitor rapamycin can alleviate the harmful effects of senescent cells in tissues and organisms by inhibiting SASP expression [19].

In this study, we aimed to explore the presence of senescent LECs in a rat model of D-galactose-induced cataracts and applied senolytics and senomorphics to determine the effects of senotherapy on cataract progression.

## 2. Methods

### 2.1. Experimental Animals

This study was approved by the Biomedical Ethics Committee of Peking University. All animal experiments complied with the ARRIVE guidelines and were conducted in accordance with the U.K. Animals (Scientific Procedures) Act, 1986, and associated guidelines. Three-week-old male Sprague–Dawley rats were purchased from the Department of Laboratory Animal Science, Peking University Health Science Center. All animals (weighing 50 ± 10 g) were kept in a specific pathogen-free facility and allowed to acclimate to this facility for 1 week before the experiment. We examined rats using a slit lamp and excluded rats with congenital cataracts. For all experiments, the rats were randomly assigned to the control or intervention groups. In the drug treatment groups, the two eyes of each rat were randomly assigned as the interventional eye and the control eye. On day 28, the rats were euthanized with an anesthetic overdose.

### 2.2. Rat Model of D-Galactose-Induced Cataracts

All rats were weighed daily and intraperitoneally injected with 10 mL/kg 50% D-galactose (Maklin, G6223, Shanghai, China) that was prepared in a 50 mL centrifuge tube (NEST Biotechnology, Cat. No. 602052, Wuxi, China). From day 10, the drinking water was replaced with 10% D-galactose water. A total of 28 days were required to establish the rat model of D-galactose-induced cataracts. On days 0, 7, 14, 21, and 28, the rat pupils were dilated with topical tropicamide (5 mg/mL)/phenylephrine (5 mg/mL) eye drops. We then examined the rats using a slit lamp (TOPCON, Tokyo, Japan) and Pentacam 70700 (OCULUS Optikgeräte GmbH, Wetzlar, Germany) to assess lens opacity.

### 2.3. Eye Drops Administration and Subconjunctival Injection

For the senolytic groups, D+Q solutions were prepared using 10% PEG400. Animals were topically treated with vehicle, D (4 μM) + Q (80 μM), D (8 μM) + Q (160 μM), or D (16 μM) + Q (320 μM) for eye drop administration and subconjunctival injection. Eye drops were maintained on the ocular surface for 1 min twice daily. From day 10, subconjunctival injections (75 μL) were administered every three days after topical anesthesia with oxybuprocaine (4 mg/mL). Before subconjunctival injections, at least three-time use of oxybuprocaine was required. To verify the effects of SASP inhibitors, we prepared eye drops containing different chemicals, including rapamycin (an mTOR inhibitor) and SB203580 (a p38 MAPK inhibitor). SASP inhibitors were dissolved in 4% DMSO, 30% PEG300, 5% Tween-80, and 61% ddH_2_O.

### 2.4. Opacity Index

Rats with dilated pupils were assessed using a slit lamp. Based on the cloudiness of the lenses, we classified them into nine grades, which were adapted from a previously published grading system [20]: stage 0, normal lenses; stage 1a, vacuoles as an equatorial ring; stage 1b, vacuolization covering more than one-third of the anterior cortex; stage 1c, vacuolization covering more than two-thirds of the anterior cortex; stage 2a, cortical opacity with some clear areas and vacuoles; stage 2b, cortical opacity with some clear areas and without vacuoles; stage 3, uniform opalescence; stage 4, nuclear opacity; and stage 5, mature opacity of the entire lens (Appendix A). Each cataract stage was assigned a score of 1 to 9 as the opacity index.

### 2.5. Lens Density

Scheimpflug imaging with the Pentacam 70700 was used to examine the anterior segment of the eye. Rats with dilated pupils were fixed in a dark room in front of the device. Within approximately 2 min, 25 images of the lenses at different angles were obtained. Along the shape of the lens, we drew a closed curve around the entire lens, and the device subsequently provided the values for average lens density (ALD), lens density standard deviation (LDSD), and maximum lens density (MLD) of the marked region in each image. The mean density value from the 25 images was regarded as the final lens density. ALD represents the opacity of the entire lens, LDSD is the lens heterogeneity, and MLD characterizes the most turbid region.

### 2.6. Hematoxylin and Eosin Staining

The rats were euthanized with 300 mg/kg 3% pentobarbital sodium. After euthanasia, the eyeballs were immediately enucleated and kept in a FAS fixation solution (Servicebio, Cat. No. G1109, Wuhan, China) for 24 h. After fixation, the eyeballs were cut sagittally to obtain the eye cups, which were embedded in paraffin after dehydration. Paraffin blocks were then sectioned at 4-μm thickness and stained with hematoxylin and eosin (H&E). H&E staining was conducted following the manufacturer’s protocol using the Hematoxylin and Eosin Staining Kit (Beyotime, Cat. No. C0105M, Shanghai, China).

### 2.7. Immunohistochemical Staining

For immunohistochemical (IHC) staining, 3% hydrogen peroxide was used to remove endogenous peroxidase activity. Antigen retrieval was performed in an EDTA buffer in a pressure cooker for 10 min. The tissues were blocked with 1% goat serum at room temperature (20–25 °C) for 20 min and incubated with primary anti-p16 (Abcam, Cat. No. ab54210, Cambridge, UK, 1:10,000) and anti-gamma H2A.X (γH2AX; Abcam, Cat. No. ab81299, Cambridge, UK, 1:250) antibodies overnight at 4 °C in a humidified chamber. Each slide was treated with a horseradish peroxidase-conjugated secondary antibody at room temperature for 20 min and reacted with diaminobenzidine (Beyotime, Cat. No. P0203, Shanghai, China) for 2 min. Finally, cell nuclei were counterstained with Mayer’s hematoxylin solution (Wako, Cat. No. 131-09665, Osaka, Japan). We randomly selected five fields for each IHC slide and took photos. The images were automatically analyzed by Image Pro Plus 6.0, which produced the values of mean optical density (MOD). The average value of five images was recorded as the expression level of one IHC slide, which was used for further statistical analyses.

### 2.8. Statistical Analysis

IBM SPSS 26.0 and GraphPad Prism 8.01 were used to analyze the data. A Q-Q plot was used to test the normality of each variable. For normally distributed data, a *t*-test and ANOVA were performed for independent variables, and a paired *t*-test was used for dependent variables. For non-normally distributed data, the Mann–Whitney U test or Kruskal–Wallis test was used for independent variables, and the Wilcoxon test was used for dependent variables. The Dunnett correction was used for multiple comparisons. Statistical significance was set at *p* < 0.05.

## 3. Results

### 3.1. Establishment of the Rat Model of D-Galactose-Induced Cataracts

Using D-galactose, we established a rat model of cataracts (Figure 1A). All rats that had received D-galactose developed vacuoles as equatorial rings at day 14, and lens transparency decreased gradually (Figure 1B). By contrast, the lenses of rats in the control group remained transparent (Figure 1B). In Scheimpflug images, the lens density increased after D-galactose administration in comparison to that of control animals (Figure 1C–E). From day 14, the opacity indices of the D-galactose-induced cataract lenses were significantly higher than those of the normal lenses (Figure 1F). On day 28, the ALD, LDSD, and MLD values of the D-galactose cataract lenses were also significantly higher than those of the normal lenses (Figure 1G). Moreover, we compared both eyes of each rat and found no differences.

The opacity index is subjective and may not distinguish well the status of lenses in adjacent categories, which makes the objective lens density a more useful parameter. Therefore, we randomly selected 12 eyes of each grade in our study and investigated the association between opacity index and objective lens density (Appendix A). For opacity indices 1 to 7, ALD and LDSD increased slowly, whereas MLD increased stepwise. For opacity indices above 7, MLD plateaued, but ALD and LDSD increased the sensitivity. Therefore, MLD was more suitable for evaluating lenses with an opacity index ≤7, whereas ALD and LDSD were more reliable for evaluating lenses with an opacity index >7.

### 3.2. Senescent Lens Epithelial Cells Accumulate in D-Galactose-Induced Cataracts

Age-related cataracts are associated with increased numbers of senescent LECs [5,21], and a previous study reported that D-galactose induces LEC senescence [10]. Therefore, we examined senescent LECs in D-galactose-induced cataracts. Compared to control lenses, fibers of lenses exposed to D-galactose were collapsed and replaced by an amorphous substance (Figure 2A,B). The structure of the lens epithelium became multilayered and disorganized. In addition, LECs became enlarged and distorted with increased vacuoles, which are regarded as morphological characteristics of senescent cells [2]. p16 and γH2AX, which are involved in CDK-associated cell cycle arrest and DNA damage response, respectively, are often used as specific markers of senescence [2]. Therefore, we examined the expression of these two markers in LECs. We observed that γH2AX^+^ and p16^+^ LECs accumulated in D-galactose-treated lenses, and the expression level of γH2AX also increased (Figure 2C,D), indicating LEC senescence. These results suggest that the number of senescent LECs is increased in D-galactose-induced cataracts.

### 3.3. D+Q Treatment Tends to Alleviate the Progression of D-Galactose-Induced Cataracts in the Early Stage

As senescent LECs accumulated in D-galactose-induced cataracts, we attempted to explore the role of these cells in cataract progression. The combination of D+Q has been proven to be an effective senolytic to eliminate senescent cells and is beneficial in many senescence-associated diseases [15,16,17]. We administered eye drops containing various D+Q concentrations to model rats (Figure 3A). In rats treated with D+Q in our study, the opacity index showed no significant differences (Figure 3B,C). ALD (Adjusted *p* value > 0.999 for low concentration; adjusted *p* value = 0.367 for moderate concentration; adjusted *p* value = 0.530 for high concentration), LDSD (Adjusted *p* value = 0.104 for low concentration; adjusted *p* value = 0.039 for moderate concentration; adjusted *p* value = 0.018 for high concentration), and MLD (Adjusted *p* value = 0.099 for low concentration; adjusted *p* value = 0.027 for moderate concentration; adjusted *p* value = 0.020 for high concentration) values were lower on day 21 after D+Q administration, which was independent of the D+Q concentration used (Figure 3D–F). However, this difference between the D+Q and vehicle groups disappeared on day 28. To exclude individual differences, we only instilled the D+Q solution into one eye of each D-galactose-treated rat and the vehicle solution into the other eye (Figure 3A). For eyes treated with a high D+Q concentration, the ALD (*p* = 0.039), LDSD (*p* = 0.039), and MLD (*p* = 0.039) values on day 21 were significantly lower than those of the corresponding control eyes. However, this was not observed in the D+Q groups with low or moderate drug concentrations (Appendix A).

Next, we performed IHC stainings for p16 and γH2AX in these lenses. Compared to vehicle-treated lenses, the expression level of γH2AX decreased gradually with the increase in D+Q concentration, especially for the high D+Q concentration (Figure 4A–E). The expression level of p16 was also slightly reduced after D+Q treatment (Appendix A). Furthermore, we compared both eyes in each rat, and no significant differences in p16 and γH2AX expression were found (Appendix A).

### 3.4. Rapamycin, a SASP Inhibitor, Tends to Alleviate the Progression of D-Galactose-Induced Cataracts in the Early Stage of Cataract

SASP-related cytokines and chemokines are secreted by senescent cells, and this can accelerate inflammation and disease progression [22]. As we noticed that the therapeutic D+Q effects disappeared on day 28, which we considered to be caused by a breakdown of the LEC barrier after the elimination of senescent LECs, we tried to inhibit SASP-associated changes. We used two SASP inhibitors (rapamycin and SB203580) to treat D-galactose-induced cataracts and found that rapamycin significantly alleviated lens opacity (Rapamycin vs. vehicle, adjusted *p* value < 0.0001) (Figure 5A). Opacity indexes, as well as ALD (Adjusted *p* value = 0.088), LDSD (Adjusted *p* value= 0.119), and MLD (Adjusted *p* value = 0.220) values of rats in the rapamycin group were lower than those in the vehicle-treated group on day 21 (Figure 5B–E). However, these effects disappeared on day 28. Another SASP inhibitor, SB203580, is a p38 inhibitor that has been shown to have a higher ALD at day 28 (Adjusted *p* value = 0.027); no statistical significances were found for opacity index, LDSD, and MLD (Figure 5A–E). Comparisons between treated and untreated eyes from each rat are displayed in Appendix A.

## 4. Discussion

Cataracts, which are related to aging, are the leading cause of vision loss worldwide. Recently, various researchers have reported that the number of senescent LECs is increased in eyes with cataracts. However, the association between senescent LECs and cataracts has rarely been investigated. Thus, in this study, we explored how the elimination of senescent LECs and SASP inhibition influence cataract progression.

Cataract is an age-related disease. In our study, we found that LECs appeared to be enlarged and distorted with increased vacuoles and elevated expression of p16 and γH2AX. This indicates increased senescence of LECs in lenses with D-galactose-induced cataracts. In addition to the D-galactose-induced cataract model, other congenital cataract models also show increased senescence of LECs, such as heat shock 4-deficient mice and vimentin-deficient mice [23,24]. Moreover, several population-based studies have reported that LEC senescence is significantly increased in cataract lenses [5,6]. Taken together, these results suggest that an increase in LEC senescence is a common feature of different types of cataracts. By damaging mitochondrial function and redox homeostasis, D-galactose can induce various types of organic aging and cellular senescence, including aging of neurons, the liver, the heart, and the reproductive system [25]. Xu et al. reported that D-galactose can accelerate LEC senescence by disturbing autophagy flux and mitochondrial functions [10].

Senolytics can eliminate senescent cells. In this study, we observed that D+Q administration reduced the LEC senescence burden and significantly alleviated cataract progression on day 21. Several studies have shown that D+Q can reduce the senescence burden, decrease the expression of SASP-related markers, and subsequently alleviate disease progression, such as ischemic nephropathy, age-dependent intervertebral disc degeneration, and insulin resistance [16,26,27]. Baker et al. reported that a transgenically induced reduction in the number of senescent cells can reduce the cataract rate in aging mice [11], which is consistent with our results. However, we also observed no difference between the lenses in the D+Q and vehicle groups on day 28. D-galactose can impair mitochondrial function, increase the osmotic pressure in the lens, influence the Na^+^/K^+^ ATPase, and damage redox homeostasis, which accelerates lens turbidity [8,9,10]. Therefore, D-galactose can induce cataracts via multiple pathways. We considered that other pathways became dominant at the late cataract stage, which rendered D+Q insufficient for inhibiting cataracts on day 28. However, this requires further investigation.

Rapamycin is an mTOR inhibitor that inhibits the expression of SASP-related cytokines and alleviates inflammation and disease progression [19]. Moreover, rapamycin has been shown to alleviate many age-related diseases [15,16,17]. In the present study, we found that rapamycin slowed cataract progression. A previous study reported that rapamycin can improve Gja8b knockout-associated autophagy impairment and subsequently relieve cataracts [28], which is consistent with our findings. However, Wilkinson et al. found that the long-term use of rapamycin slowed down the aging of various organs but increased the occurrence of age-related cataracts [29]. Thus, the optimal dose and duration of rapamycin treatment require further investigation. SB203580 prevents SASPs by inhibiting the p38 MAPK pathway. Unexpectedly, we found that SB203580 significantly promoted cataract formation, in contrast to previous research findings [30,31]. Inhibition of p38 MAPK can reduce H_2_O_2_-induced LEC apoptosis [31], which may help alleviate cataracts. We considered that these contradictory findings were caused by different concentrations of the p38 MAPK inhibitors.

Previous studies have shown that Scheimpflug imaging is reliable for evaluating lens transparency in animal and human lenses [32,33]. However, no prior study has reported the application range of Scheimpflug imaging in animals. When the opacity index reached 8 in this study, the MLD and LDSD values remained stable, whereas the ALD values changed significantly. For opacity indexes below 8, MLD and LDSD values were sensitive to index changes, whereas ALD values increased only slightly. These results indicate that ALD is a more reliable parameter at the late stage of cataracts, whereas MLD and LDSD are much more sensitive at the early stage of cataracts.

This is the first study to apply the senolytic D+Q to cataracts and observe the therapeutic effects of senescence-targeting therapy on cataract progression, which provides a potential way to postpone the occurrence and progression of cataracts. However, our study has some limitations. First, we performed experiments using the D-galactose-induced cataract model, which is more similar to a metabolic cataract rather than an age-related cataract. D-galactose-induced cataracts always initially represented as the opacity in the cortex, and gradually developed into the nuclear opacity or thorough opacity. However, classic age-related cataracts initially presented with increased nuclear density, subsequently followed by nuclear opacity or cortical opacity. In fact, nuclear cataract is the most common type for age-related cataracts, which is different from D-galactose-induced cataracts. Even for cortical cataracts, the age-related cataract is wheel-like cloudy, while the opacity pattern of D-galactose-induced cataracts we observed in our study was irregular. This may limit the clinical application of senolytic agents. Second, we did not explore the mechanisms by which senescent LECs induce cataracts. Third, although we found that rapamycin can alleviate cataract progression, we did not explore the suitable drug dosage, duration of therapy, and drug delivery method, which requires further investigation.

In conclusion, senescent LECs were responsible for D-galactose-induced cataracts. Senolytic D+Q alleviated D-galactose-induced cataract progression by reducing the senescent LEC burden in the early stage of cataract. The SASP inhibitor rapamycin tends to slow down cataract formation in the early stage of D-galactose-induced cataracts. However, further investigations to confirm this phenomenon and explore the underlying mechanisms are still needed.

## Figures and Tables

**Figure 1 jfb-14-00006-f001:**
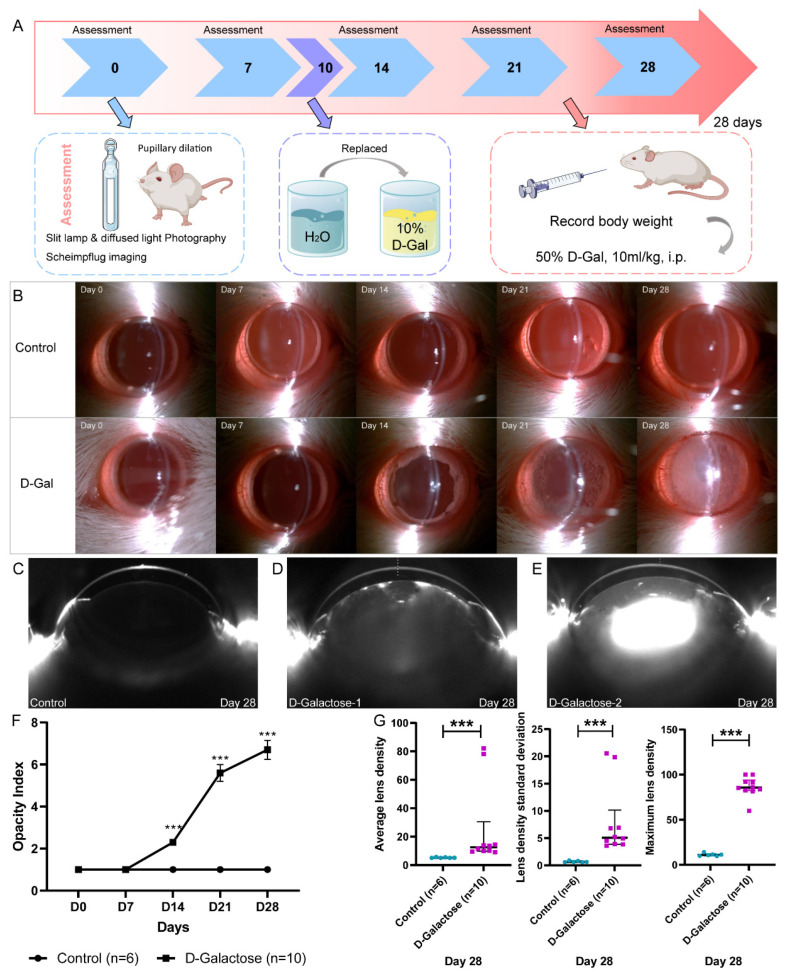
Establishment of the rat model of D-galactose-induced cataracts. (**A**), Flowchart of cataract induction with D-galactose. (**B**), Anterior segment photographs of control and D-galactose-induced cataract lenses. (**C**), Scheimpflug images of normal lenses. (**D**,**E**), Scheimpflug images of cataract lenses. (**F**), Changes in opacity index. (**G**). Changes in average lens density, lens density standard deviation, and maximum lens density. ***, *p* < 0.001; D-gal, D-galactose; i.p., intraperitoneal.

**Figure 2 jfb-14-00006-f002:**
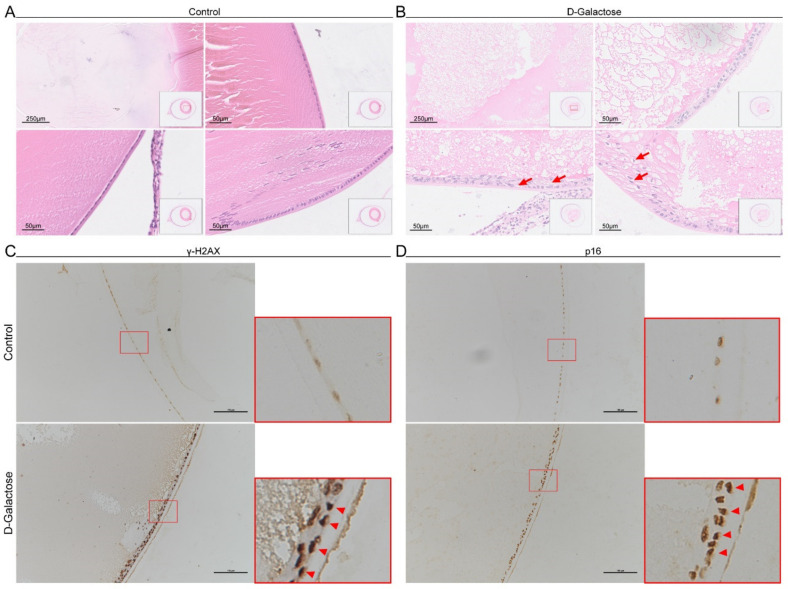
Senescent lens epithelial cells accumulate in cataract lenses. (**A**), H&E staining of control lenses. (**B**), H&E staining of D-galactose-induced cataract lenses; the arrows indicate the vacuolized lens epithelial cells. (**C**), IHC staining for γH2AX of control and cataract lenses. (**D**), IHC staining for p16 of control and cataract lenses. The arrowheads indicate the γH2AX- or p16-upregulated lens epithelial cells. H&E, hematoxylin and eosin; IHC, immunohistochemical; γH2AX, gamma H2A.X.

**Figure 3 jfb-14-00006-f003:**
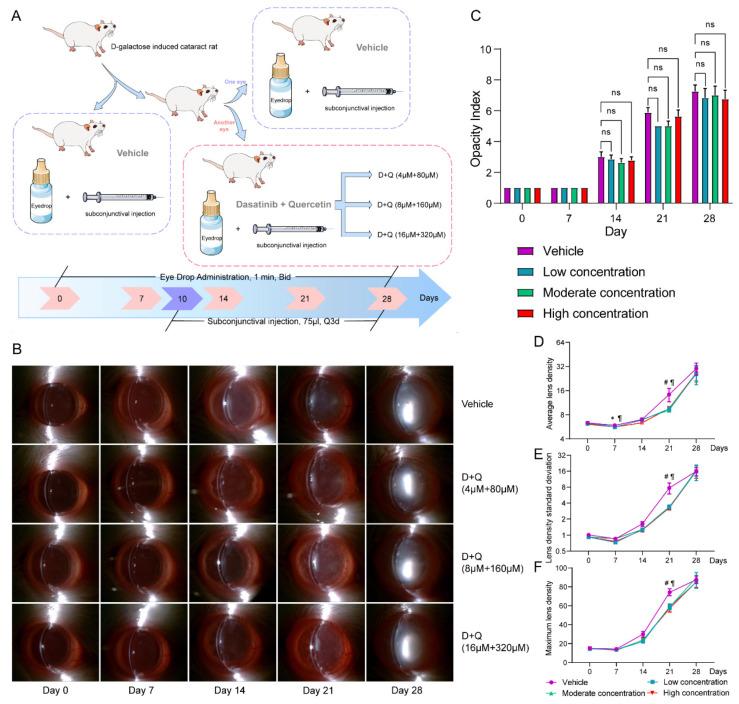
D+Q alleviate cataract progression at the early stage. (**A**), Flowchart of the D+Q administration. (**B**), Anterior segment photographs of cataract lenses in the D+Q and vehicle groups. (**C**), Changes in opacity index in the D+Q and vehicle groups. (**D**–**F**), Changes in average lens density, lens density standard deviation, and maximum lens density in the D+Q and vehicle groups. *, vehicle vs. low D+Q concentration (*p* < 0.05); #, vehicle vs. moderate D+Q concentration (*p* < 0.05); ¶, vehicle vs. high D+Q concentration (*p* < 0.05); ns, not significant; D+Q, dasatinib plus quercetin.

**Figure 4 jfb-14-00006-f004:**
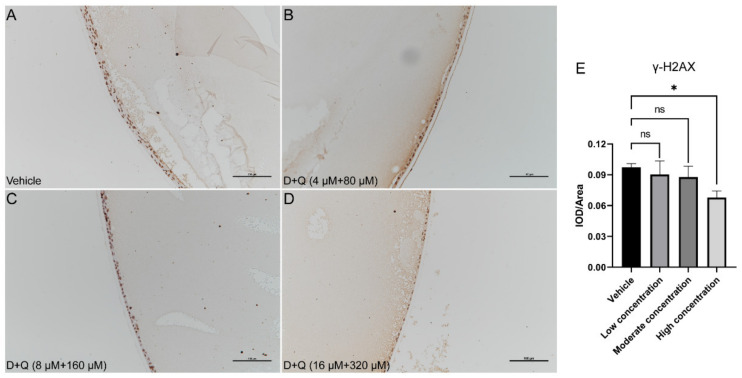
The γH2AX expression in cataract LECs is reduced after D+Q administration. (**A**–**D**), IHC staining for γH2AX in cataract lenses after administration of D+Q or vehicle. (**E**), Expression levels of γH2AX in cataract LECs after D+Q or vehicle administration. D+Q, dasatinib plus quercetin; IHC, immunohistochemical; LEC, lens epithelial cell; ns, not significant; γH2AX, gamma H2A.X; *, *p* < 0.05.

**Figure 5 jfb-14-00006-f005:**
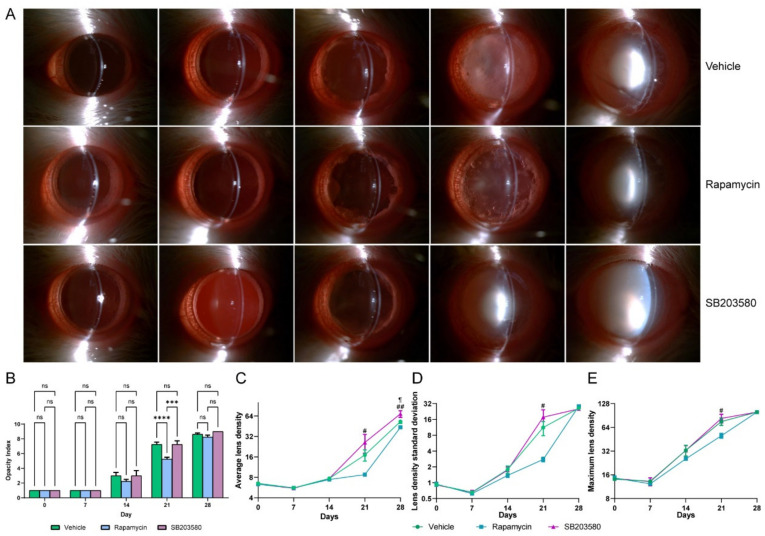
Rapamycin alleviates the cataract progression, while SB203580 aggravates the cataract. (**A**), Anterior segment photographs of cataract lenses after administration of rapamycin, SB203580, or vehicle. (**B**), Changes in opacity index after administration of rapamycin, SB203580, or vehicle. (**C**–**E**), Changes in average lens density, lens density standard deviation, and maximum lens density after administration of rapamycin, SB203580, or vehicle. ns, not significant; ***, *p* < 0.001; ****, *p* < 0.0001; ¶, vehicle vs. SB203580 (*p* < 0.05); #, rapamycin vs. SB203580 (*p* < 0.05); ##, rapamycin vs. SB203580 (*p* < 0.01). D+Q, dasatinib plus quercetin; IHC, immunohistochemical; LEC, lens epithelial cell; γH2AX, gamma H2A.X.

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
