# Peer review of "Reduction in Lens Epithelial Cell Senescence Burden through Dasatinib Plus Quercetin or Rapamycin Alleviates D-Galactose-Induced Cataract Progression"

_jfb, 2022, doi:10.3390/jfb14010006_

Round 1
Reviewer 1 Report
1. The authors can improved the background by added the burden of disease with epidemiological data.
2.Authors also can add the aim of study at the end of background.
3. Authors also can update the references not more than 10 years.
4. Authors can add the methods, how to anesthetized the rats before treatment.
5.Authors can add the methods, how to euthanized the rats before enucleasion eyeball.
Author Response
1. The authors can improve the background by added the burden of disease with epidemiological data.
Answer 1: Thank you for your valuable suggestions. We have added the burden of cataract with epidemiological data in the introduction as follows: “Cataract is an age-related disease that is one of the leading global causes of blind-ness or moderate/severe vision impairment (MSVI) in those aged 50 years and older in 2020 (15.2 million cases [95% uncertainty intervals = 12.7-18.0] for blindness, 78·8 million cases [67·2-91·4] for MSVI)”.
2. Authors also can add the aim of study at the end of background.
Answer 2: We appreciate the reviewer’s comment. The aim of the study was presented at the end of background: “In this study, we aimed to explore the presence of senescent LECs in a rat model of D-galactose-induced cataracts and apply senolytics and senomorphics to determine the effects of senotherapy on cataract progression”.
3. Authors also can update the references not more than 10 years.
Answer 3: Thanks for the reviewer’s kind remind. We have updated the references.
4. Authors can add the methods, how to anesthetized the rats before treatment.
Answer 4: We treated the rats through eyedrops or subconjunctival injections. We grasped and controlled the rat, subsequently applied the eyedrops, and hold on for 1 minutes. This process did not required anesthesia. When performing subconjunctival injections, topical anesthesia of oxybuprocaine was used. The detail of this procedure was added in the methods.
5. Authors can add the methods, how to euthanized the rats before enucleating eyeball.
Answer 5: We have added the procedure of euthanasia before eyeball enucleation in the methods. Thank you for your suggestion.
Reviewer 2 Report
Thank you the for the opportunity the manuscript by Wang et al. In this manuscript the authors explored senescence in a D-galactose-induced cataract model in rats. While the work the authors did is important, the way a rather small difference at 21 days is presented, in my opinion is misleading. I still do think the data is worth to be published in this journal.
Introduction:
In the introduction, only senescence is introduced. Cataract itself is only broadly mentioned. Please add a few sentences on the rat model you used and how it relates to other rat models of cataract progression.
Results:
The authors describe that in D-galactose induced cataract there are more senescent cells. As the authors point out, this is not a new fact.
“We observed that γH2AX+ and 166 p16+ LECs accumulated in D-galactose-treated lenses, and the expression level of γH2AX 167 was also increased (Fig. 2C-D), indicating LEC senescence. “
How is expression quantified? This does not only apply to this analysis but to the following as well:
“Compared to 199 vehicle-treated lenses, the expression level of γH2AX decreased gradually with the in- 200 crease in D+Q concentration, especially for the high D+Q concentration (Fig. 4A-E) “
Page 5, last paragraph
I think the results of this paragraph do not justify its heading. The authors do not mention adjusting for multiple testing, they do not mention p values and one small, most likely clinically insignificant difference in ALD; LDSD and MLD cannot be considered to alleviate cataract progression. I do believe the results are interesting, but this needs to be rephrased.
Page 7:
Again, I do not believe the paragraph heading is supported by the data presented. Again, no p-values and no adjustment for multiple testing is mentioned.
Discussion:
In my opinion, the discussion needs to be rewritten.
Discussing the difference between Scheimpflug imaging and the rather subjective opacity index is not the main message of this paper and should not be the first paragraph of the discussion.
D-Galactose is not a classic cataract as the authors pointed out. I would have liked the discussion to elaborate the difference in morphology between galactose-cataracts and classic age-related cataracts further. This is important with an eye towards translating the work the authors did into clinical work.
Again, I would be cautious to say that this study showed that any of the agents used slowed cataract progression as absolutely no difference was to be seen at 28 days.
In a revised version, please elaborate multiple testing adjustments and clear statistics. It is crucial for the integrity of the paper.
Author Response
1. In the introduction, only senescence is introduced. Cataract itself is only broadly mentioned. Please add a few sentences on the rat model you used and how it relates to other rat models of cataract progression.
Answer 1: Thanks for the reviewer’s kind remind. We have added the information of the rat model we used, and compared it with other cataract models in the induction.
2. The authors describe that in D-galactose induced cataract there are more senescent cells. As the authors point out, this is not a new fact. “We observed that γH2AX+ and p16+ LECs accumulated in D-galactose-treated lenses, and the expression level of γH2AX was also increased (Fig. 2C-D), indicating LEC senescence.”
Answer 2: Good point. The previous study found that the D-galactose could induced the LEC senescence, which was based on the ex vivo study. In our study, we validated this in vivo, which we think is a new fact.
3. How is expression quantified? This does not only apply to this analysis but to the following as well: “Compared to vehicle-treated lenses, the expression level of γH2AX decreased gradually with the increase in D+Q concentration, especially for the high D+Q concentration (Fig. 4A-E) “
Answer 3: Thank you for your question. We are so sorry that we did not clearly clarify the qualification process. We randomly selected five fields for each IHC slide and took photos. The images were automatically analyzed by Image Pro Plus 6.0, which produced the values of mean optical density (MOD). The average value of five images was recorded as the expression level of one IHC slide, which was used for further statistical analyses. This process was added in the methods.
4. Page 5, last paragraph. I think the results of this paragraph do not justify its heading. The authors do not mention adjusting for multiple testing, they do not mention p values and one small, most likely clinically insignificant difference in ALD; LDSD and MLD cannot be considered to alleviate cataract progression. I do believe the results are interesting, but this needs to be rephrased.
Answer 4: Thank you for the reviewer’s suggestion. We have rephrased our expression, and added the P value and adjusted P value after adjusting multiple comparison.
5. Page 7: Again, I do not believe the paragraph heading is supported by the data presented. Again, no p-values and no adjustment for multiple testing is mentioned.
Answer 5: We have rephrased our expression, and added the adjusted P value after adjusting multiple comparison.
6. Discussing the difference between Scheimpflug imaging and the rather subjective opacity index is not the main message of this paper and should not be the first paragraph of the discussion.
Answer 6: We appreciate the reviewer’s constructive suggestion. We agree that this part is not the main message of this paper, and we have transferred this paragraph to the later part of discussion.
7. D-Galactose is not a classic cataract as the authors pointed out. I would have liked the discussion to elaborate the difference in morphology between galactose-cataracts and classic age-related cataracts further. This is important with an eye towards translating the work the authors did into clinical work.
Answer 7: Thank you for your comment. We have compared the classic age-related cataract with galactose cataract in the discussion. “D-galactose induced cataract always initially represented as the opacity in the cortex, and gradually progressed till the nuclear opacity or thoroughly opacity. However, clas-sic age-related cataract initially presented with increased nuclear density, subsequent-ly followed by nuclear opacity or cortical opacity. In fact, nuclear cataract is the most common type for age-related cataract, which is different from D-galactose induced cat-aract. Even for cortical cataract, the age-related cataract is wheel-like cloudy, while the opacity pattern of D-galactose induced cataract we observed in our study was irregular”.
8. Again, I would be cautious to say that this study showed that any of the agents used slowed cataract progression as absolutely no difference was to be seen at 28 days.
Answer 8: We are sorry for our unprecise expression. We have rephrased our words in the discussion as follows: “In conclusion, senescent LECs were responsible for D-galactose-induced cataracts. Senolytic D+Q alleviated D-galactose-induced cataract progression by reducing the se-nescent LEC burden in the early stage of cataract. The SASP inhibitor rapamycin tends to slow down cataract formation in the early stage of cataract. However, further investigations to confirm this phenomenon and explore the underlying mechanisms are still needed”.
9. In a revised version, please elaborate multiple testing adjustments and clear statistics. It is crucial for the integrity of the paper.
Answer 9: We have elaborated multiple testing adjustments and statistics in the revised version, and we hope this version is clear enough to read. The Dunnett correction was used in our paper.
Reviewer 3 Report
The manuscript studied the effect of three treatment regimens dasatinib plus quercetin (D+Q) and senescence-associated secretory phenotype (SASP) inhibitors, rapamycin and SB203580 on lens epithelial cell senescence in D-galactose-induced cataract progression. The effect of treatments were assessed by the opacity index, the average lens density (ALD), lens density standard deviation (LDSD), and maximum lens density (MLD). Additionally, histological studies and Immunohistochemical stainings for markers of senescence p16 and γH2AX were performed.
My comments
-Treatments used should be indicated in the title
-Information about various treatments used should be added to the introduction (Origin: natural or chemical, mechanisms of action reported in senescence, examples of senescence-related diseases reported)
- In histological data, changes observed on figures should be indicated on figures as arrows, arrow-heads, etc
-The manuscript appears as two combined manuscripts in context of results, why the authors separate dasatinib plus quercetin (D+Q) from senescence-associated secretory phenotype inhibitors. ALso, H&E and IHC were not applied for groups treated with senescence-associated secretory phenotype inhibitors. Same for summary of experiemntal procedure
-Treatment sometimes indicated as dasatinib plus quercetin or D+Q, please unify
-Figure 5. Rapamycin alleviates the cataract progression: Should mention/comment on SB203580 to avoid confusion
-The study did not indicate or even clearly discuss the possible underlying mechanisms of results reported
Author Response
1. Treatments used should be indicated in the title
Answer: Thanks for the reviewer’s kind remind. We have edited the title as the reviewer suggested as follows: “Reduction in lens epithelial cell senescence burden through dasatinib plus quercetin alleviates D-galactose induced cataract progression”.
2. Information about various treatments used should be added to the introduction (Origin: natural or chemical, mechanisms of action reported in senescence, examples of senescence-related diseases reported)
Answer: We appreciate the reviewer’s valuable suggestions. We have added the information about the treatments we used in the introduction.
3. In histological data, changes observed on figures should be indicated on figures as arrows, arrow-heads, etc
Answer: We have indicated the histological changes observed on figures. Thanks for the reviewer’s suggestion.
4. The manuscript appears as two combined manuscripts in context of results, why the authors separate dasatinib plus quercetin (D+Q) from senescence-associated secretory phenotype inhibitors. Also, H&E and IHC were not applied for groups treated with senescence-associated secretory phenotype inhibitors. Same for summary of experimental procedure.
Answer: Thank you very much for your comment. The D+Q and SASP inhibitors functions through the different mechanisms in theory. Besides, we decided to use the SASP inhibitors due to the results of D+Q. We assumed that the therapeutic effect of D+Q disappeared at day 28 was possibly due to the elimination of senescent LECs and subsequent broken of lens epithelial barrier. Thus, we tried SASP inhibitor to inhibit the functions of senescent LECs without breaking the lens epithelial barrier. Therefore, we separate D+Q from SASP inhibitors. We are sorry that we did not conduct the H&E and IHC for the groups treated with SASP inhibitors. We hoped that the drug can be used in the clinical in the future, so we mainly focused on the treatment effect for cataract rather than the histological changes. Therefore, the eyeballs in the groups treated with SASP inhibitors were nor preserved.
5. Treatment sometimes indicated as dasatinib plus quercetin or D+Q, please unify
Answer: We appreciated the reviewer’s kind remind, and we have unified this problem.
6. Figure 5. Rapamycin alleviates the cataract progression: Should mention/comment on SB203580 to avoid confusion.
Answer: We have revised the figure legends, and thanked the reviewer’s remind.
7. The study did not indicate or even clearly discuss the possible underlying mechanisms of results reported
Answer: Thank you for your comment. We agreed that the underlying mechanisms help us better understand the cataract progression and design treatment targets. In our study, we focused on the results of phenotype, lacking exploring the underlying mechanisms. In the future, we will conduct experiments to make up this part.
Round 2
Reviewer 2 Report
“Another SASP 253 inhibitor, SB203580, is a p38 inhibitor that has been shown to slightly accelerate cataract 254 progression, although without statistical significance (Fig. 5A-E) “
Please rephrase. Slightly is imprecise wording. Either it accelerates cataract or not (based on p-value and statistics). If there is no statistical difference, there is no need the quantify the relevance of an assumed difference in this setting.
“In conclusion, senescent LECs were responsible for D-galactose-induced cataracts. 338 Senolytic D+Q alleviated D-galactose-induced cataract progression by reducing the senes- 339 cent LEC burden in the early stage of cataract.
Please add „in the early stage of D-galactose induced cataract“.
I thank the authors for their explanations on statistics.
Author Response
1. “Another SASP inhibitor, SB203580, is a p38 inhibitor that has been shown to slightly accelerate cataract progression, although without statistical significance (Fig. 5A-E)”. Please rephrase. Slightly is imprecise wording. Either it accelerates cataract or not (based on p-value and statistics). If there is no statistical difference, there is no need the quantify the relevance of an assumed difference in this setting.
Answer: Thank you very much for your kind remind, we have rephrased this sentence as follows: “Another SASP inhibitor, SB203580, is a p38 inhibitor that has been shown to have a higher ALD at day 28 (Adjusted P value = 0.027), no statistical significances were found for opaci-ty index, LDSD, and MLD (Fig. 5A-E)”.
2. “In conclusion, senescent LECs were responsible for D-galactose-induced cataracts. Senolytic D+Q alleviated D-galactose-induced cataract progression by reducing the senescent LEC burden in the early stage of cataract.” Please add “in the early stage of D-galactose induced cataract”.
Answer: We have added these words in the sentence.
Reviewer 3 Report
The authors addressed my comments
Minor change
The use of SASP inhibitor rapamycin has not been indicated in the title
Author Response
The use of SASP inhibitor rapamycin has not been indicated in the title
Answer: We have added the rapamycin in the title.